# Combining Ability Analysis of Yield-Related Traits of Two Elite Rice Restorer Lines in Chinese Hybrid Rice

**DOI:** 10.3390/ijms241512395

**Published:** 2023-08-03

**Authors:** Shiguang Wang, Haoxiang Wu, Zhanhua Lu, Wei Liu, Xiaofei Wang, Zhiqiang Fang, Xiuying He

**Affiliations:** 1Rice Research Institute, Guangdong Academy of Agricultural Sciences, Guangzhou 510640, China; wangshiguang@gdaas.cn (S.W.); wuhaoxiang@gdaas.cn (H.W.); luzhanhua@gdaas.cn (Z.L.); liuwei@gdaas.cn (W.L.); wangxiaofei@gdaas.cn (X.W.); fangzhiqiang@gdaas.cn (Z.F.); 2Key Laboratory of Genetics and Breeding of High Quality Rice in Southern China (Co-Construction by Ministry and Province), Ministry of Agriculture and Rural Affairs, Guangzhou 510640, China; 3Guangdong Key Laboratory of New Technology in Rice Breeding, Guangzhou 510640, China; 4Guangdong Rice Engineering Laboratory, Guangzhou 510640, China

**Keywords:** hybrid rice, Yuenongsimiao, restorer of fertility, GCA, SCA

## Abstract

Hybrid rice breeding is an important strategy for enhancing grain yield. Breeding high-performance parental lines and identifying combining abilities is a top priority for hybrid breeding. Yuenongsimiao (YNSM) and its derivative variety Yuehesimiao (YHSM) are elite restorer lines with a high ability of fertility restoration, from which 67 derived hybrid combinations have been authorized to different degrees in more than 110 instances in China. In this study, we found that YNSM and YHSM contained three candidate restorer-of-fertility (*Rf*) genes, *Rf3*, *Rf4*, and *Rf5*/*Rf1a*, that might confer their restoration ability. Subsequently, we investigated heterosis and combining ability of YNSM and YHSM using 50 F_1_ hybrids from a 5 × 10 incomplete diallelic mating design. Our results indicated that hybrid combinations exhibited significant genetic differences, and the additive effects of the parental genes played a preponderant role in the inheritance of observed traits. The metrics of plant height (PH), 1000-grain weight (TGW), panicle length (PL), and the number of spikelets per panicle (NSP) were mainly affected by genetic inheritance with higher heritability. Notably, the general combining ability (GCA) of YHSM exhibited the largest positive effect on the number of grains per panicle (NGP), NSP, PL, and TGW. Thus, YHSM had the largest GCA effect on yield per plant (YPP). In addition, the GCA of YNSM exhibited a positive impact on YPP, mainly due to the critical contribution of seed setting percentage (SSP). Moreover, YNSM and YHSM exhibited negative GCA effects on PH, implying that YNSM and YHSM could effectively enhance plant lodging resistance by reducing the plant height of the derived hybrids. Remarkably, among the hybrids, Yuanxiang A/YNSM (YXA/YNSM), Shen 08S/Yuemeizhan (S08S/YMZ), and Quan 9311A/YHSM (Q9311A/YHSM) represent promising new combinations with a higher specific combining ability (SCA) effect value on YPP with a value more than 3.50. Our research thus highlights the promising application for the rational utilization of YNSM and YHSM in hybrid rice breeding.

## 1. Introduction

Rice is one of the most important staple cereals for global food security, and there is still great potential to elevate rice grain yields to address the needs of growing populations [1,2,3,4]. To date, rice breeding has experienced four significant stages of improvement in the forms of dwarf breeding, heterosis utilization, green super rice (GSR) cultivation, and rational design breeding in order to achieve a steady increase in rice grain yield [5,6,7,8,9,10,11]. Since the 1970s, hybrid rice has successfully led to a second quantum leap in rice yields through the exploitation and utilization of heterosis [12,13].

Previous rice breeding practices verified that heterosis utilization combined with morphological optimization is an effective approach for breeding super-high-yield rice varieties [13,14,15]. Meanwhile, the three-line or cytoplasmic male sterility (CMS) system and two-line or thermo-sensitive/photoperiod-sensitive genic male sterile (T/PGMS) system have been most successfully applied in hybrid rice breeding to utilize heterosis [13,16]. It is well known that the breeding and rational selection of excellent parental lines is the prerequisite to the successful development of elite hybrids. For instance, Shuhui527 (R527), an elite backbone parent of hybrid rice, has been widely applied in hybrid rice breeding programs and used for the derivative creation of other excellent restorer lines [17]. Huazhan (HZ) is an elite restorer line with excellent characteristics, high yield potential, good grain quality, wide adaptability, and wide affinity. To date, 158 hybrid rice combinations derived from HZ have been registered and cultivated widely in China [18]. Previous studies indicate that the *indica*/*japonica* intersubspecific hybrid combinations exhibit higher heterosis strength and grain yield potential [13]. Shuhui498 (R498) is an elite parent of heavy panicle hybrid rice and has been successfully used to develop a large number of super-high-yield heavy panicle-type intersubspecific hybrid rice varieties [19,20]. R9308 was the descendent of an intersubspecies cross and carried about a quarter of *japonica* genetic components; its derived super hybrid rice combination Xieyou 9308 exhibited super-high-yielding capacity with ideotype and strong hybrid vigor [21]. Conversely, C418, a *japonica*-type wide-compatibility restorer line with higher *indica* genetic components, showed the characteristics of high yield and high combining ability [11]. To capitalize on intersubspecific heterosis and overcome the reproductive barrier of *indica*-*japonica* hybrids, the wide-compatibility (*Wc*) gene has been widely used for breeding many outstanding *indica*-compatible *japonica* parental lines, such as restorer lines 9311, E32, 1826, R1128, and TGMS line Pei’ai 64S (PA64S), etc. [11,22].

As mentioned above, the highly efficient strategy for developing super-high-yielding hybrids is to improve further and breed parental backbone lines with excellent agronomic characteristics, high combining ability, and broad-spectrum compatibility [23,24]. The two-line or T/PGMS system possesses the advantage of the free combination of parents [16]. Correspondingly, applying the three-line or CMS system requires the combination of a male sterile line, a restorer line, and a maintainer line, and its success depends mainly on the restorer line [13,25]. Remarkably, a large number of elite restorer lines, such as Guanghui998 (R998), Guanghui308 (R308), and Guanghui122 (R122), were developed and are widely utilized, and outstanding achievements and breakthroughs have been made in hybrid rice breeding by Rice Research Institute of Guangdong Academy of Agricultural Sciences [26,27]. 

Notably, variety-restorer line rice breeding has made significant achievements and played an important role in rice production in South China. YNSM and its derived variety YHSM are excellent conventional rice varieties with excellent comprehensive characteristics of high quality, high and stable yield, high harvest index, disease resistance, and wide adaptability. As such, these lines have been characterized as national standard second-grade rice varieties and selected as the leading agricultural varieties in Guangdong Province for consecutive years [28]. Pointing toward the wide adaptability of these two varieties, YNSM has been successfully registered and released for commercial production in Guangdong, Hainan, Guangxi, and Hubei Provinces and had been passed the introduction application for Hunan, Jiangxi, Anhui, Henan, and Fujian Provinces. Meanwhile, YHSM has been successively registered and released for commercial production in Guangdong, Jiangxi, Hubei, and Anhui Provinces and has passed the introduction application for Henan and Zhejiang Provinces. Meanwhile, YNSM and YHSM are also used as elite restorer lines with a high fertility restoration ability for hybrid rice breeding, of which 67 derived hybrid combinations have been authorized at different levels over 110 times in China (from China Rice Data Center). Despite the utilization of these traits, the reasons for the high fertility restoration and combining abilities of YNSM and YHSM remain elusive. In this study, we identified several types of *Rf* genes using relevant specific markers to investigate the reasons for the high fertility restoration abilities of YNSM and YHSM. Subsequently, we generated 50 F_1_ hybrids to evaluate the heterosis and combining ability of YNSM and YHSM using the incomplete diallelic hybridization 5 × 10 (NCⅡ). Our results showed that YHSM exhibited the largest GCA effect on YPP and found that Yuanxiang A/YNSM was probably promising new combinations with the highest SCA effect value of YPP. Our research thus highlights the high efficiency of variety-restorer line rice breeding and shows a promising application of YNSM and YHSM.

## 2. Results

### 2.1. YNSM and YHSM Are High-Performance Restorer Lines for Hybrid Rice

As elite restore lines, YHSM and YNSM exhibit a high fertility restoration ability for combining multiple P/TGMS and CMS sterile lines (Appendix A). To date, 67 derivative hybrid rice combinations of YNSM and YHSM have been authorized more than 110 times in different rice ecological regions, such as the middle-lower reaches of the Yangtze River, the upper reaches of the Yangtze River, and Southern China (Appendix A). The prominent usage of these genetic resources prompted us to identify the distribution of several types of *Rf* genes in these two restorer lines. The results showed that similar to the elite restorer lines R527, HZ, and R998, both YNSM and YHSM carried the three *Rf* genes, *Rf3*, *Rf4*, and *Rf5*/*Rf1a*, but did not carry the *Rf6* gene (Figure 1). Thus, YNSM and YHSM might exhibit high fertility restoration ability via the activity of these genes, which prompted us further to evaluate the combining ability of YNSM and YHSM through a series of crosses and scoring of progenies through yield-related traits.

### 2.2. Variance Analysis of Combining Ability

The coefficients of variation (CV) of the agronomic traits of the hybrids and their restorer lines are summarized in Appendix A. The agronomic traits of 50 F_1_ hybrids possessed a higher CV than that of their parents except for PL, which displayed a higher degree of variation (Appendix A), which suggested that these agronomic traits of the hybrids exhibited a higher degree of variation. The comparison of yield components showed that the NSP of hybrids was superior to that of its restorer lines, while the YPP was inferior to that of its restorer lines due to a lower SSP (Appendix A). Notably, the SSP and NGP of hybrids derived from YHSM and YNSM were superior to that of HZ and R998, thereby resulting in higher YPP (Figure 2). On the other hand, the CV of the yield components of the hybrids derived from YHSM and YNSM were less than those of the HZ and R998 (Figure 2). These results indicated that the hybrids derived from YHSM and YNSM exhibited superior and stable yield performance. Secondly, we performed variance and combining ability analyses and found that there were significant differences (*p* < 0.01) in yield-related traits among hybrids, indicating that there were significant genetic differences between hybrids (Table 1). Similar to previously reported results [22,29,30,31,32], PH, grain number (NSP and NGP), and YPP showed significant differences (*p* < 0.05 or *p* < 0.01) among the replicates in our present study, suggesting that these four traits were susceptible to environmental influence (Table 1). In addition, the variance in the SCA (female × male) for the studied traits was significantly less than that for GCA (parents), indicating that the performance of these traits was determined by both additive and non-additive effects, with additive effects dominating in their performance outcomes (Table 1). These findings were further supported by the higher variance contributing rate of GCA (Table 2), which also suggested that the GCA of parents might play a more significant role than that of SCA in selecting for heterosis in rice breeding.

### 2.3. Genetic Effect Analysis of Combining Ability

Subsequently, we estimated the parental variance contribution rate of particular genetic parameters to yield correlation data of characteristics for a better understanding of the influence of parental genetics on offspring heterosis. Our results showed that the proportion of variance in GCA (Vg) of the eight agronomic traits contributing to the total variance was higher than that in SCA (Vs), accounting for 66.00–99.35% of the variation. This indicated that the parental gene additive effect played a leading role in the performance of the hybrid generation (Table 2). Interestingly, the proportion of female and male variance (Vg1 and Vg2, respectively) in the variance of Vg varies according to agronomic traits. In this study, the Vg1 of all the investigated traits was larger than Vg2. Only the Vg2 of EPN was slightly different from Vg1, indicating that the GCA of the sterile lines played a dominant role in the performance of these traits (Table 2).

Meanwhile, the proportion of Vs in TGW, EPN, NGP, and YPP to the total variance was more than 30%, suggesting that the role of parental interaction of the sterile line and restorer line was indispensable for these traits (Table 2). In addition, the analysis of the rate of contribution of genotype variance to yield-related traits showed that the environment greatly affected the four traits of EPN, SSP, NGP, and YPP (Table 2). Heritability is the percentage of the genetic variation in a given trait in the total variation. Our results exhibited that the broad heritability (hB2) was higher comparatively than the narrow heritability (hN2) for all of the studied traits, which suggested that the additive variance in the parental trait was the primary component of the total genotypic variance (Table 2). In addition, our results demonstrated that PH, TGW, PL, and NSP were mainly affected by genetic inheritance with a high heritability of more than 50%, which can be selected for in early generations. Conversely, the hN2 of EPN and YPP was relatively low, under 30%, which implied that the inheritance of these traits was unstable and susceptible to the environment and gene non-additive effects (Table 2).

### 2.4. YNSM and YHSM Exhibited High Combining Ability for Yield-Related Traits

Combining ability is an important index to measure the utilization value of parents, which determines the strength of heterosis. Analysis of GCA for yield-related traits in parents revealed significant differences in GCA between parents and among other traits of the same parent (Table 3 and Appendix A). For restorer lines, the GCA of YHSM exhibited a positive and significant effect on grain number (NSP and NGP), PL, and TGW among all studied restorer lines (Table 3). Therefore, YHSM had the most significant GCA effect (0.98) on YPP to effectively increase the yield of its hybrid combinations (Table 3). In addition, the GCA of YMZ and YNSM exhibited a positive effect, while the GCA of HZ and R998 negatively impacted YPP. The effect of YMZ on YPP was mainly due to the importance of EPN, and therefore, SSP played a significant role in the effect of YNSM on YPP (Table 3).

In comparison, the GCA effects of YHSM and YNSM on NGP and SSP were higher than that of HZ and R998, reaching more than 8.50 and 3.00, respectively, resulting in the GCA effect of YHSM and YNSM on YPP being higher than that of HZ and R998 (Table 3). Notably, both YHSM and YNSM exhibited negative GCA effects on PH, which were −2.85 and −2.15, respectively, implying that YHSM and YNSM effectively reduced the plant height in the resultant hybrid combinations to enhance plant lodging resistance, which may have great potential application in future lodging-resistant breeding of rice (Table 3). Considering the sterile lines, Quan 9311A and Fudao A showed positive effects on all monitored yield-related traits except EPN and SSP, respectively, and consequently exhibited higher GCA effects of more than 4.00 on YPP (Appendix A), indicating that Quan 9311A and Fudao A are promising candidates for breeding high-yield rice varieties with large panicles and more spikelets. On the other hand, Guang 8A and Yuanxiang A showed significantly negative effects on TGW (Appendix A), which could be potentially utilized in breeding high-quality rice varieties with slender grains.

The SCA effect value is one of the critical indicators to evaluate the superiority and inferiority of the hybrid combinations. The SCA effect values of 50 F_1_ hybrids are presented in Appendix A. There were significant differences in SCA effects between traits of the same combination and combinations of the same character. Specifically, the SCA effect of YPP ranged from −4.40 to 7.99, among which the SCA effect of the combinations YXA/YNSM, S08S/YMZ, and Q9311A/YHSM was higher than 3.50; the SCA effect value of EPN of the combinations Shu 2A/YMZ and G8A/YMZ was the greatest and least, respectively; the SCA effect value of NSP of the combinations Tai S/YMZ and YXA/YMZ was the greatest and least, respectively; the SCA effect value of NGP of the combinations YXA/YNSM and Q9311A/YMZ was the greatest and least, respectively; the SCA effect value of SSP of the combinations WFA/YMZ and Wo S/YNSM was the greatest and least, respectively; the SCA effect value of TGW of the combinations Q9311A/YMZ and Tai S/YMZ was the greatest and least, respectively; the SCA effect value of PL of the combinations S08S/YMZ and S08S/YHSM was the greatest and least, respectively; the SCA effects of PH ranged from −7.14 to 4.92, and the SCA of the combination S08S/R998 was mostly negative effect (Appendix A). Integrally, the combination of YXA/YNSM exhibited the highest SCA effect value of YPP and more prominent grain yield performance, followed by the two combinations of S08S/YMZ and Q9311A/YHSM (Appendix A). In addition, the SCA effects of yield-related traits such as EPN, PL, NSP, NGP, SSP, and TGW of YXA/YNSM showed relatively higher and positive effects (Appendix A). These results suggested that YXA/YNSM, S08S/YMZ, and Q9311A/YHSM are promising new hybrid combinations, but overall, YXA/YNSM was the optimal hybrid combination for yield-related traits.

### 2.5. Correlation Analysis of Yield Components

To further analyze the influencing factors of high rice yield, we performed a correlation analysis between the yield components of the hybrid combinations with positive SCA of YPP and all 50 F_1_ hybrid combinations, respectively. Statistical analysis showed that there were significant (*p* < 0.01) and positive correlations between grain number (NSP and NGP) and YPP, and NGP exhibited the highest positive correlation with YPP (Figure 3). Notably, SSP had a significant correlation with grain number but no significant correlation with YPP (Figure 3). The EPN and TGW exhibited significant (*p* < 0.01 and *p* < 0.05, respectively) and positive correlations with YPP in all of the 50 F_1_ hybrid combinations (Figure 3B), while there were no significant correlations between EPN and YPP or TGW and YPP in the hybrid combinations with positive SCA of YPP, among which, the EPN was negatively correlated with YPP (Figure 3A). In addition, PL exhibited significant (*p* < 0.01) and positive correlations with grain number and YPP, while SSP exhibited no significant and negative correlations with EPN and TGW (Figure 3A). In summary, these results suggested that grain number was the critical factor for determining the high yield of rice in these hybrid varieties, and EPN, SSP, and TGW were the prominent plastic factors that affected the total yield of rice. Therefore, our primary breeding goal should be cultivating large panicle hybrid rice combinations with long panicles and a large number of grains to improve the rice yield further.

## 3. Discussion

Hybrid rice, including three-line and two-line hybrids, which exhibit yield advantages of 10−20% higher than inbred rice varieties, have been commercially released and distributed worldwide since the 1970s [24]. The three-line hybrid rice system mainly includes several types, such as WA-CMS, HL-CMS, and BT-CMS [25], while there was no restriction for the two-line hybrid system regarding the restorer line. In contrast, restorer lines must carry specific nuclear *Rf* genes for a given CMS sterile line [25]. For instance, *Rf3* and *Rf4* restore the fertility of WA-CMS sterile lines, *Rf5*/*Rf1a* restores the fertility of BT-CMS and HL-CMS sterile lines, and *Rf6* restores the fertility of HL-CMS sterile lines, etc. Improvement and breeding of excellent parental lines are the keys to the success of breeding elite hybrids. As excellent parental lines, NSM and YHSM have been widely utilized as seed cores in rice breeding [28]. In our present study, we demonstrated that both YNSM and YHSM carried three *Rf* genes, *Rf3*, *Rf4*, and *Rf5*/*Rf1a*, which is a similar feature to the elite restorer lines R527, HZ, and R998 (Figure 2). Previous studies revealed that these three major types of CMS systems, including WA-CMS, HL-CMS, and BT-CMS, contributed to about 60% of the global hybrid rice production [33,34]. Specifically, approximately 99% of hybrids have been derived from the WA-CMS/*Rf* system in rice [35]. Thus, it was reasonable to infer that YNSM and YHSM exhibited a high ability for fertility restoration may be due to the inherent combination of *Rf* genes. 

Combining ability is a vital parameter to measure the utilization value of parents, which is widely used in screening elite parents and improving breeding efficiency in hybrid breeding programs [36,37,38,39]. Practically speaking, an effective strategy for developing potential hybrids with strong heterosis is breeding excellent parents with high GCA effects combined with strong SCA. In the practice of hybrid rice breeding, it is frequently observed that a series of elite hybrid combinations have been developed from a single superior parent line, such as R527, HZ, and R998, etc. [17,18,26]. In this study, YHSM and YNSM exhibited a higher and more positive GCA effect on YPP than HZ and R998 due to the differences in GCA effects on NGP and SSP (Table 3). Additionally, the yield performance of hybrid rice combinations derived from YHSM and YNSM was superior to that of HZ and R998, with a lower CV of yield components, indicating more reliable traits in the progeny (Figure 2). These results suggested that YHSM and YNSM might possess the characteristics of producing hybrid rice combinations with higher and more stable yields. On the other hand, these results indicated that the SSP played a significant role in the GCA effect of YPP and the yield performance of derived hybrid rice combinations of YHSM and YNSM, which further demonstrated the characteristics of YNSM and its derivatives with favorable late ripening color and high seed setting rate in the late mature stage [28]. 

In a hybridization program, the estimation of heterosis and combining abilities is a powerful tool to clarify the nature of gene action for desirable traits, which is also helpful in evaluating the genetic potential of parent lines for subsequent selection [30,31]. GCA reflects additive gene actions and is directly related to the breeding value of parents and is theoretically fixable. In contrast, SCA reflects non-additive gene activity resulting from dominance, overdominance, and epistatic effects, which impact the performance of hybrids and are non-fixable [29,31,40]. Our genetic effect and variance analysis results showed that all traits studied in this work were determined by additive and non-additive gene actions, with additive effects dominating in all traits, implying that the parent lines exhibited high breeding values with strong potential for heterosis (Table 1 and Table 2). These results corroborate those of previous studies [31,32,41]. Deep investigation of the heterosis of particular F_1_ hybrids still requires further commercial usage. Heterosis or hybrid vigor is the superior performance of hybrids compared to that of their parents. This ability has been widely used in multiple crops that lead to a tremendous increasement in yields [42]. If the heterosis is predominantly determined by additive gene actions and completely consistent with the dominance hypothesis, it is theoretically possible to select homozygous dominant individuals and fix the heterosis to develop superior inbred lines. However, it is unlikely to achieve that goal in practice due to genetic linkage, gene interactions, and the accumulation of minor effect genes. Three major hypotheses of dominance, overdominance, and epistasis have been proposed to explain heterosis [43]. Additionally, the additive and dominant allele-specific expression genes (ASEGs) play important roles in heterosis, and the patterns of CG methylation in gene bodies play crucial roles in heterosis, as revealed by a recent study [44]. Recent results imply that the ratio of differentially methylated regions in CG context in exons to transcription start sites between the parents could be a feasible predictor for heterosis potential, as the ratio was significantly negatively correlated with the magnitude of heterosis of their hybrids [44]. Therefore, intensive efforts to better characterize the contributions of various molecular mechanisms of heterosis remain a focus for future research.

Exploring the high combining ability of desired traits is necessary for maximizing heterosis for a given purpose, such as maximizing grain yield. Previous studies indicate that the negative heterosis and combining ability effect for plant height was desirable for breeding short-statured varieties and hybrids of rice to avoid lodging [24,29,30,32,41]. Notably, YNSM and YHSM showed negative GCA effects for plant height, thereby implying their potentiality for developing short-statured hybrids to enhance lodging resistance (Table 3). The Green Revolution gene *semidwarf* 1 (*sd1*) can effectively modulate the semi-dwarf phenotype to enhance lodging resistance and improve the harvest index of rice [5,6,7]. Remarkably, *sd1* significantly contributes to GCA effects with additive effects on plant height in hybrids and probably influences mid-parent heterosis and over-standard heterosis [45,46]. Our previous results showed that YNSM and YHSM carry the null allele of *sd1*, which conferred exhibited excellent lodging resistance and a high harvest index [28,47]. Thus, we deduced that the GCA effect of YNSM and YHSM on plant height might be due to their elusive inherent genetic basis, which warrants further investigation. Specifically, the YXA/YNSM, S08S/YMZ, and Q9311A/YHSM combinations were promising new combinations with higher SCA effect values of YPP and prominent grain yield performance (Appendix A). Our results thus suggested that YNSM and YHSM were excellent restorer lines with a high combining ability for the breeding of superior hybrids.

The first hybrid combination was developed in 1974, and hybrid rice was commercialized in 1976 [13]. Subsequently, Chinese hybrid rice experienced six rounds of variety renovation from 1985 to 2015, while the major high-quality hybrid rice varieties in China have been updated one to two times in the recent ten-odd years (data from the National Agro-Tech Extension and Service Center of China, NATESC). The major inspiration for the renovation of hybrid rice varieties was disease resistance and quality improvement. Recently, hybrid rice has become less competitive, and its planting area has gradually decreased to about 45% due to the continuous improvement in the yield and quality of conventional rice varieties [13,26]. Therefore, breeding disease-resistant and high-quality parental lines is an efficient strategy for developing potential hybrid rice varieties. YNSM and YHSM exhibited excellent comprehensive characteristics such as high and stable yield, high quality, high resistance to rice blast, and medium resistance to bacterial blight [28]. Remarkably, the derivative hybrids of YNSM and YHSM possessed good disease resistance and produced high-quality rice grain (Appendix A). For instance, the hybrid combination Woliangyou YHSM showed high resistance to rice blast and resistance to bacterial blight, and its quality reached the first-grade standard of high-quality rice issued by the Ministry of Agriculture and Rural Affairs (MARA) of China (Appendix A). Meanwhile, Jingliangyou 1212 and Longliangyou 1212, authorized as super rice varieties by MARA with the second-grade standard of high-quality rice (Appendix A), are currently being widely promoted and cultivated in large areas.

Shanyou63, derived from the parents Zhenshan97A and Minghui63, is an elite mega hybrid rice variety due to its high yield and wide adaptability with weak photoperiod sensitivity. It has been cultivated geographically in 16 provinces from 18° N (Hainan) to 38° N (Shandong) [48]. Jingliangyou–Huazhan (JLYHZ) and Longliangyou–Huazhan (LLYHZ), which are derived from the identical restorer line HZ, exhibited wide adaptability and rapidly became the top three widely promoted hybrid rice varieties in China [18]. Notably, YNSM and its derivative varieties, such as Jingliangyou 1212, Longliangyou 1212, Guangbayou YHSM, and Hengfengyou YHSM, etc., exhibited wide adaptability and succeeded in various ecoregional trials, which were certificated and approved for commercial rice geographic production ranging from the upper to middle-lower reaches of the Yangtze River region and Hainan to Henan geographically (Appendix A). Previous studies demonstrated that heading date is a determinant of yield and heterosis in hybrid rice, and the combinations of several major heading date genes, such as *Ghd7*, *Ghd8*, and *Hd1*, play important roles in enhancing adaptation to ecological regions in rice [49]. Notably, YNSM carried a strong functional allele of *Ghd7* and non-functional alleles of *Ghd8* and *Hd1*, which are identical to the combination of MH63 and allowed the perception of gradual changes in day length [49]. These results suggested that the derived hybrid varieties of YNSM probably have a Shanyou63-like photoperiod module which allows these varieties to achieve multi-latitude adaptation and productivity [50]. Interestingly, gradual day-length sensing coupled with optimum cropping modes enhanced multi-latitude adaptation, expanded planting areas, and improved unit yield of crops [50]. Therefore, YNSM should be used as an important backbone germplasm for future breeding potential in widely adapted varieties.

In summary, YNSM and YHSM are not only outstanding conventional rice varieties with excellent comprehensive characteristics but also excellent parents of hybrid rice with high fertility restoration and combining ability. To ensure food security and sustainable development of agriculture, developing resource-saving and environmentally friendly GSR with high yield and improved quality is the primary goal of rice breeding in the future [9,10]. The homogenization of hybrid rice is becoming increasingly severe due to an extraordinarily narrow genetic diversity of available and well-studied parental lines [17]. Thus, as elite cultivars and important parents of hybrid rice, YNSM and YHSM can serve as effective tools to efficiently contribute to the improvement in available germplasm and genetic diversity to generate GSR varieties. 

## 4. Conclusions

This study identified the presence of several different *Rf* genes in YNSM and YHSM, which indicated their potential as genetic resources for high fertility restoration ability in rice breeding efforts. Secondly, we estimated various genetic parameters, GCA, and SCA to evaluate the heterosis and combining ability of YNSM and YHSM. Taken together, we inferred that the reasons for the high fertility restoration ability of YNSM and YHSM might be that these parents pyramid three *Rf* genes, *Rf3*, *Rf4*, and *Rf5*/*Rf1a*. From our data, we found that the reason for the enhancement in GCA on yield caused by YHSM was due to the highest GCA of several yield-related traits, such as NGP, NSP, PL, and TGW. Meanwhile, the negative GCA effect on PH derived from YNSM and YHSM could provide a useful genetic resource for lodging resistance breeding in hybrid rice, which is extremely helpful for practical rice breeding. Finally, the results of analyzing SCA effects of yield-related traits illustrated three promising new combinations for high-yield rice varieties for future commercialization: YXA/YNSM, SS/YMZ, and Q9311A/YHSM. Taken together, this work explored the genetic potential of YNSM and YHSM in hybrid rice breeding and proposed promising avenues for future hybrid rice breeding efforts using these high-performing parental lines.

## 5. Materials and Methods

### 5.1. Plant Materials and Growth Conditions

Five rice restorer lines, YNSM and its derived varieties YHSM, YMZ, R998, and HZ, were used to cross with ten different types of sterile lines, including three P/TGMS lines, Tai S, Wo S, and Shen 08S, and seven CMS lines, Guang 8A, Guangtai A, Wufeng A, Quan 9311A, Fudao A, Yuanxiang A, and Shu 2A. The pedigree details of the parental lines are illustrated in Appendix A and Appendix A. We developed 50 hybrids following the incomplete diallel hybridization 5 × 10 (NCⅡ) in the later-cropping season of 2018. All hybrid combinations and restore lines were evaluated and arranged in a randomized complete block design (RCBD) in triplicate in the early cropping season of 2019 using standard field management practices at the experimental fields of Guangdong Academy of Agricultural Sciences (GDAAS), Guangzhou, Guangdong Province, China. Seeds were sown after completing germination on February 28. Then, thirty-day-old seedlings were transplanted in the main field on March 30. All materials were planted in three rows, with six plants per line in one row. The planting density was 16.7 × 16.7 cm.

### 5.2. PCR-Based Genotyping

Plant genomic DNA was extracted from fresh leaves of heading-stage plants using a modified CTAB method [51]. The molecular markers for genotyping *Rf3* were the SSR markers RM10353, RM10338, and PSM354 [52,53]; SSR marker RM6100 and the indel marker M19280 for *Rf4* [54,55]; indel marker Indel-*Rf1a* for *Rf5*/*Rf1a* [56]; and indel marker ID200-1 for *Rf6* [34]. PCR amplification was performed on a Bio-Rad C1000 Touch Thermal Cycler (USA) using a modified protocol with appropriate parameters [57]. The PCR products for the marker RM10353, RM10338, RM6100, and M19280 were resolved on an 8% polyacrylamide gel, the PCR product for the marker PSM354 was resolved on a 3.0% agarose gel, and the products for the InDel-Rf1a and ID200-1 markers were resolved on a 1.0% agarose gel in 1× TBE buffer. The primers used in this study and the PCR amplicon size of the corresponding primer are listed in Appendix A.

### 5.3. Agronomic Traits Evaluation

The 50 F_1_ hybrid combinations and 5 restorer lines were planted in an RCBD with 3 replicates in this study. At maturity, six plants per plot were harvested 35 days after heading during June and July of 2019 and air-dried for a total of three biological replicates for agronomic traits evaluation. An array of agronomic traits, including PH (cm), PL (cm), EPN, NGP, NSP, SSP (%), TGW (g), and YPP (g), were investigated for the 5 restorer lines and 50 F_1_ hybrid combinations according to a method described previously [19].

### 5.4. Statistical Analysis

Mean phenotypic values comparison, analysis of variance (ANOVA), combining ability analysis including GCA variance effects of the parents, SCA variance effects of the hybrids, variance contribution rate, and heritability were performed and calculated using Microsoft Excel 2013 and SPSS 22.0 using previously described methods [31].

## Figures and Tables

**Figure 1 ijms-24-12395-f001:**
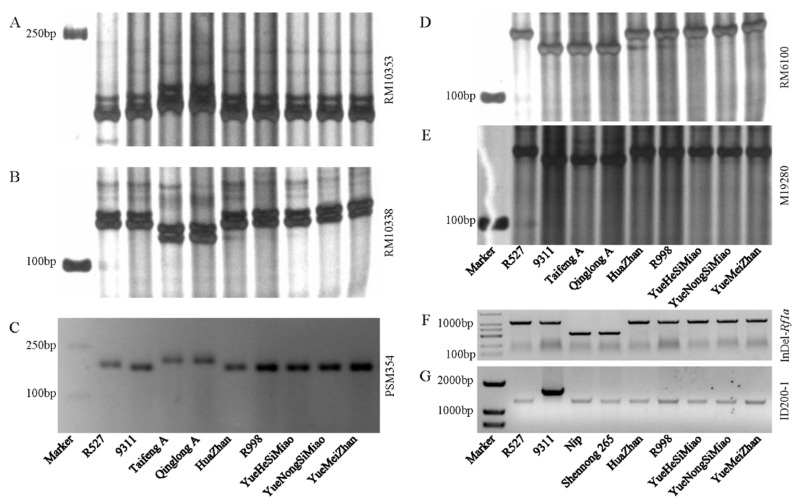
Genotype identification of restorer-of-fertility (*Rf*) genes in different parental lines. (**A**–**C**) Genotype identification of the *Rf3* gene using the simple sequence repeat (SSR) markers RM10353, RM10338, and PSM354, respectively. R527 and 9311 were the positive controls, and Taifeng A and Qinglong A were the negative controls. (**D**,**E**) Genotype identification of the *Rf4* gene using the SSR marker RM6100 and Indel marker M19280, respectively. R527 was the positive control, and 9311, Taifeng A, and Qinglong A were the negative controls. (**A**–**E**) All samples in the lanes from left to right of gel images were identical to those labeled below (**C**,**E**). (**F**) Genotype identification of the *Rf5*/*Rf1a* gene using the Indel marker InDel-*Rf1a*. R527 and 9311 were the positive controls, and Nipponbare (Nip) and Shennong 265 were the negative controls. (**G**) Genotype identification of the *Rf6* gene using the Indel marker ID200-1. Sample 9311 was the positive control, and R527, Nip, and Shennong 265 were the negative controls. (**F**,**G**) All samples in the lanes from left to right of gel images were identical to those labeled below (**G**).

**Figure 2 ijms-24-12395-f002:**
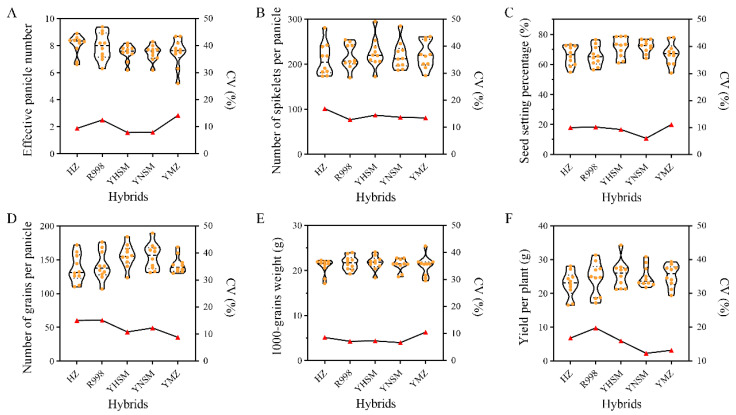
Comparison of yield components between the hybrids of each restorer line. (**A**–**F**) Statistical analysis of effective panicle number (**A**), number of spikelets per panicle (**B**), seed setting percentage (**C**), number of grains per panicle (**D**), 1000-grain weight (**E**), and yield per plant (**F**) in hybrids derived from the specific restorer line. The violin plots with value plots and the red triangles represent the phenotype and variation in the corresponding traits of these hybrids, respectively. The left and right longitudinal axes in the figure represent the value of phenotype and variation, respectively.

**Figure 3 ijms-24-12395-f003:**
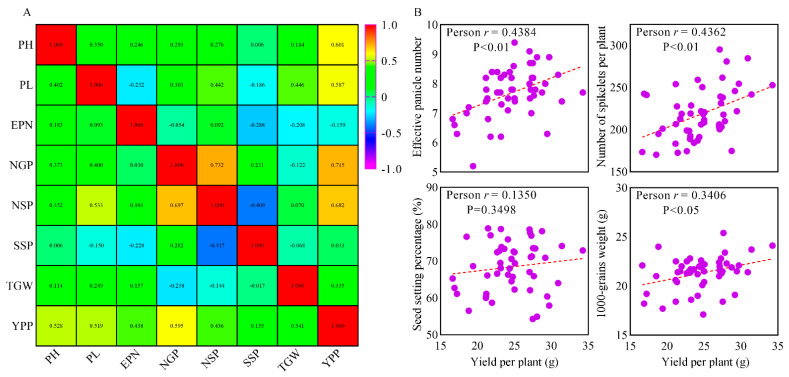
Correlation analysis of the agronomic traits. (**A**) Correlations between the eight agronomic traits. The correlation coefficient of the 25 hybrid combinations with positive SCA of YPP is in the upper triangle, and all 50 hybrid combinations are in the lower triangle. PH = plant height, PL = panicle length, EPN = effective panicle number, NGP = number grains per panicle, NSP = number spikelets per panicle, SSP = seed setting percentage, TGW = 1000-grain weight, YPP = yield per plant. (**B**) Scatter plots of effective panicle number, number of spikelets per panicle, seed setting percentage, and 1000-grain weight versus yield per plant of the 50 hybrid combinations, respectively.

**Table 1 ijms-24-12395-t001:** Variance analysis of agronomic traits.

Source of Variation	PH	PL	EPN	NGP	NSP	SSP	TGW	YPP	df
Replicates	322.74 **	0.48	0.62	2249.52 **	3977.11 **	43.61	0.06	61.83 *	2
Combinations	88.75 **	2.38 **	2.06	1110.85 **	2768.20 **	137.24 **	8.64 **	44.74 **	49
Female	306.62 **	9.74 **	3.03	2643.93 **	11,536.69 **	408.06 **	33.27 **	124.45 **	9
Male	204.70 **	1.91 *	3.83	2132.88 *	807.77	282.56 **	4.27	38.79	4
Female × Male	21.39 **	0.59	1.62	614.03 **	793.9 *	53.39 *	2.96 **	25.47 *	36
Error	6.52	0.57	1.39	326.68	439.17	33.60	0.03	15.42	98

Note: * and ** mean significantly different from the control at *p* < 0.05 and *p* < 0.01, respectively. PH = plant height, PL = panicle length, EPN = effective panicle number, NGP = number of grains per panicle, NSP = number of spikelets per panicle, SSP = seed setting percentage, TGW = 1000-grain weight, YPP = yield per plant, df = degree of freedom.

**Table 2 ijms-24-12395-t002:** Parental variance contribution rate and genetic rate analysis.

Traits	Genotype Variance				Variance Contributing Rate (%)			Heritability (%)		
	Female	Male	Female × Male	Error	Vg-GCA	Vs-SCA	Vg1	Vg2	Broad Heritability	Narrow Heritability
PH	19.015	6.110	4.959	6.518	83.52	16.48	63.21	20.31	82.19	68.64
TGW	2.021	0.044	0.976	0.034	67.91	32.09	66.47	1.44	98.89	67.15
EPN	0.094	0.074	0.077	1.391	68.67	31.33	38.55	30.12	14.94	10.26
PL	0.610	0.044	0.004	0.573	99.35	0.65	92.67	6.68	53.47	53.13
SSP	23.645	7.639	6.597	33.598	82.59	17.41	62.42	20.17	53.00	43.77
NGP	135.327	50.629	95.783	326.679	66.00	34.00	48.03	17.97	46.31	30.56
NSP	716.186	0.462	118.243	439.169	85.84	14.16	85.78	0.06	65.53	56.25
YPP	6.599	0.444	3.352	15.416	67.76	32.24	63.49	4.27	40.27	27.29

Note: Vg1 represents the proportion of female variance in the variance of general combining ability, and Vg2 represents the proportion of male variance in the variance of general combining ability.

**Table 3 ijms-24-12395-t003:** GCA effect values of agronomic traits of each restorer line.

Restorer Lines	PH	PL	EPN	NGP	NSP	SSP	TGW	YPP
HZ	−0.35	0.19	0.21	−9.26	−7.81	−2.54	−0.44	−1.89
R998	2.56	−0.40	0.12	−4.90	−1.05	−2.71	0.36	−0.17
YHSM	−2.85	0.24	−0.36	9.38	6.41	3.05	0.40	0.98
YNSM	−2.15	−0.04	−0.39	8.52	0.36	3.57	−0.04	0.41
YMZ	2.80	0.01	0.42	−3.74	2.09	−1.36	−0.29	0.66

PH = plant height, PL = panicle length, EPN = effective panicle number, NGP = number grains per panicle, NSP = number spikelets per panicle, SSP = seed setting percentage, TGW = 1000-grain weight, YPP = yield per plant, HZ = Huazhan, R998 = Guamghui 998, YHSM = Yuehesimiao, YNSM = Yuenongsimaio, YMZ = Yuemeizhan.

## Data Availability

Not applicable.

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
