# Peer review of "Combining Ability Analysis of Yield-Related Traits of Two Elite Rice Restorer Lines in Chinese Hybrid Rice"

_ijms, 2023, doi:10.3390/ijms241512395_

Round 1
Reviewer 1 Report (New Reviewer)
The authors have improved the manuscript and the additions/revisions are important for understanding the results presented.
Points that need further revisions. The results are specific to the environments that the lines were developed since they have not been tested in other environments. The title is very general and does not convey a scientific meaning. Please revise to something like :
"Combining ability analysis of yield related traits of elite restorer lines Yuenongsimiao in Chinese hybrid rice".
This is a only a suggestion, the title needs revision.
Please add a clear objective(s) and the reasons that you undertook this research study.
The authors mention that: "Results indicated that hybrid combinations exhibited significant genetic differences and the additive gene action played a preponderant role in the inheritance of studied traits"
Since additive gene action plays a preponderant role in the inheritance of most of the genetic traits in rice, rice inbred lines can be developed that are as high yielding as hybrid rice. In the case of rice, it is very expensive to develop hybrids because rice is a self pollinated crop with a predominance of additive gene action. The authors do not mention anything about the significant amount of money, effort and labor that goes into developing rice hybrids as compared to inbred lines. Please add a separate section addressing these concerns.
Please add a separate section summarizing the Conclusions of your research.
English language is overall good, some minor editing is needed since certain sentences do not read well.
Author Response
Point 1: The results are specific to the environments that the lines were developed since they have not been tested in other environments. Response 1: We thank the reviewer for pointing out this issue. We indeed should have applied tests for our hybrids and restore lines in othe environments. In present study, all hybrids and restore lines were planted in a randomized complete block design with three replicates. At maturity, six plants per plot were harvested and air-dried as three biological replicates for agronomic traits evaluation. Therefore, our experimental design involves both technical and biological duplication, ensuring the accuracy and reliability of experimental data. Point 2: The title is very general and does not convey a scientific meaning. Please revise to something like : "Combining ability analysis of yield related traits of elite restorer lines Yuenongsimiao in Chinese hybrid rice". This is a only a suggestion, the title needs revision. Response 2: Thank you for your valuable comments and suggestions. We have made appropriate revision to the title of the paper in the revised manuscript based on your suggestion. Point 3: Please add a clear objective(s) and the reasons that you undertook this research study. Response 3: Thank you for your valuable comments. We have followed your suggestion to add the reasons why we implemented this study in the Introduction section of the revised manuscript. Point 4: The authors mention that: "Results indicated that hybrid combinations exhibited significant genetic differences and the additive gene action played a preponderant role in the inheritance of studied traits". Since additive gene action plays a preponderant role in the inheritance of most of the genetic traits in rice, rice inbred lines can be developed that are as high yielding as hybrid rice. In the case of rice, it is very expensive to develop hybrids because rice is a self pollinated crop with a predominance of additive gene action. The authors do not mention anything about the significant amount of money, effort and labor that goes into developing rice hybrids as compared to inbred lines. Please add a separate section addressing these concerns. Response 4: Thank you for your valuable comments. We quite agree with your opinion. General combining ability (GCA) is directly related to the breeding value of a parent and is associated with additive gene effects, while specific combining ability (SCA) is the relative performance of a cross that is associated with non-additive gene action, predominantly contributed by dominance, epistasis, or genotype-environment interaction effects. For maximizing heterosis expression, the necessary is to explore the high combining ability of desired traits. In a hybridization program, estimation of heterosis and combining abilities is a powerful tool to clarify the nature of gene action for desirable traits, which is also helpful to evaluate the genetic potential of parent lines for subsequent selection. The goal of the diallel analysis is to divide the data’s total variation into GCA and SCA of parents and crosses, respectively. It also explains how the quantity of additive and non-additive gene action affects parents’ ability to generate superior offspring. In our present study, the variance of the SCA (female×male) for studies traits was significantly less than that of GCA (parents), indicating that the performance of these traits was determined by both additive and non-additive effects, with additive effects dominating (Table 1). It was further supported by the higher variance contributing rate of GCA, indicating that the parental gene additive effect played a leading role in the performance of the hybrid generation (Table 2). In additon, our results exhibited that the broad heritability (hB2) was higher comparatively than the narrow heritability (hN2) for all the studied traits, which suggested that the additive variance was the primary component of the total genotypic variance (Table 2). However, the hN2 of EPN and YPP was low, less than 30%, which implied that the inheritance of these traits was unstable and susceptible to the environment and gene non-additive effects (Table 2). These results revealed that all studied traits were determined by additive and non-additive gene actions, while the additive variance was the preponderance of the total genotypic variance. We have followed your suggestions to add this conclusion to the Discussion section addressing your concerns. Point 5: Please add a separate section summarizing the Conclusions of your research. Response 5: Thank you for your valuable comments. We have added a separate section to summarize our conclusions in the revised manuscript. Point 6: English language is overall good, some minor editing is needed since certain sentences do not read well. Response 6: Thank you for your valuable and thoughtful comments. We have carefully checked and improved the English writing in the revised manuscript. We really hope that the language level have been substantially improved.
Reviewer 2 Report (New Reviewer)
Dear authors. Congratulations on successfully completing a breeding study on a crucial crop. Your study has some outstanding results. However, there are some issues that need to be improved. My suggestions are marked in the PF file. Some of my concerns are as follows:
- Please use proper sentences and coordination for each section. In the abstract, there is a result sentence written before the material and method sentence. In the Results, there are a lot of discussion sentences when you already have a separate Discussion section. The results also include some material and methods sentences. Please clarify all sections in a proper flow.
- The English expression used in the manuscript is not bad. However, you can use more fluent and direct sentences. I suggested some corrections in the PDF.
-Tables and figures must be self-explanatory. even though the figures include relevant information, tables lack the details. Please write the full version of the abbreviated traits as a footnote.
- Please highlight some of the superior hybrid lines.
- Please abbreviate the traits when they first appear and use the abbreviated version throughout the manuscript.

English quality is enough, but some improvements can be performed to obtain an easily readable manuscript. My suggestions are in the PDF file.
Author Response
Point 1: Please use proper sentences and coordination for each section. In the abstract, there is a result sentence written before the material and method sentence. In the Results, there are a lot of discussion sentences when you already have a separate Discussion section. The results also include some material and methods sentences. Please clarify all sections in a proper flow. Response 1: Thank you for your valuable and thoughtful comments. We apologize for the confusion generated by the previous manuscript. We have followed your sincere suggestions to carefully check and revise all sections to ensure using proper sentences and coordination for each section in the revised manuscript. Point 2: The English expression used in the manuscript is not bad. However, you can use more fluent and direct sentences. I suggested some corrections in the PDF. Response 2: Thank you for your valuable comments and suggestions. We have polished our paper based on your useful suggestions to improve the fluency and readability of the sentences in the revised manuscript. Point 3: Tables and figures must be self-explanatory. even though the figures include relevant information, tables lack the details. Please write the full version of the abbreviated traits as a footnote. Response 3: Thank you for your valuable comments. We have followed your suggestions to add the full version of the abbreviated traits as a footnote in corresponding figures and tables. Point 4: Please highlight some of the superior hybrid lines. Response 4: Thank you for your valuable comments. We have highlighted some of the superior hybrids in our manuscript. For instance, the hybrids Woliangyou YHSM showed high resistance to rice blast and resistance to bacterial blight with the first-grade standard of high-quality; Jingliangyou 1212, Longliangyou 1212, Guangbayou YHSM, and Hengfengyou YHSM, etc, exhibited wide adaptability and got through various ecoregional trials. Among them, Jingliangyou 1212 and Longliangyou 1212, authorized as super rice varieties by the Ministry of Agriculture and Rural Affairs of China with the second-grade standard of high-quality. Point 5: Please abbreviate the traits when they first appear and use the abbreviated version throughout the manuscript. Response 5: Thank you for your valuable comments. We have carefully checked and worked on the text to correct this issue in the revised manuscript. Point 6: English quality is enough, but some improvements can be performed to obtain an easily readable manuscript. My suggestions are in the PDF file. Response 6: Thank you for your useful suggestions. We have worked on language in the revised manuscript based on your useful suggestions. We really hope thae the language level and readability have been substantially improved.
Round 2
Reviewer 1 Report (New Reviewer)
The authors have made improvements, however there are still problems with the accuracy of data.
“The 50 F1 hybrid combinations and five restorer lines were planted in a randomized complete block design with three replicates in the present study. At maturity, six plants per plot were harvested and air-dried as three biological replicates for agronomic traits evaluation. An array of agronomic traits including PH (cm), PL (cm), EPN, NGP, NSP, SSP (%),TGW (g) and YPP (g) were investigated for the five restorer lines and 50 F1 hybrid combinations.”
Authors replied to my comments as follows:
“We thank the reviewer for pointing out this issue. We indeed should have applied tests for our hybrids and restore lines in the environments. In present study, all hybrids and restore lines were planted in a randomized complete block design with three replicates. At maturity, six plants per plot were harvested and air-dried as three biological replicates for agronomic traits evaluation. Therefore, our experimental design involves both technical and biological duplication, ensuring the accuracy and reliability of experimental data.”
As I mentioned, data on agronomic polygenic traits that are based only in one environment and 3 replications are simply not accurate. I disagree that this experimental design ensures the accuracy and reliability of experimental data and I believe that any statistician or plant breeder will comment that 3 replications are not enough to make valid conclusions. You can only make preliminary conclusions till you apply your tests in other environments as well.
Furthermore, there are is no information on the experimental trial. Please state the location and year the experiment was planted. What was the row length and plant density? What was the unit of evaluation, how many rows did you plant per entry?
In addition, the authors did not respond to my comments on the amount of money, effort and labor that goes into the development of hybrids. To my knowledge, it is cheaper and more productive to spend this amount of time and money to develop superior inbred lines in rice.
In the case of rice, it is very expensive to develop hybrids because rice is a self pollinated crop with a predominance of additive gene action, meaning that inbred lines can be developed that are as high yielding as hybrid rice. The authors do not mention anything about the significant amount of money, effort and labor that goes into developing rice hybrids as compared to inbred lines. Please add a separate section addressing these important concerns and why you believe that developing hybrids is a cost-effective solution to rice breeding.
Due to the aforementioned problems, I believe that this paper may be published as a note rather than a regular article.
English language is good
Author Response
Point 1: As I mentioned, data on agronomic polygenic traits that are based only in one environment and 3 replications are simply not accurate. I disagree that this experimental design ensures the accuracy and reliability of experimental data and I believe that any statistician or plant breeder will comment that 3 replications are not enough to make valid conclusions. You can only make preliminary conclusions till you apply your tests in other environments as well.
Response 1: Thank you for your scientific and rigorous comments, which will be of great benefit to our future research. In our present study, all hybrids and restore lines were evaluated and arranged in a randomized complete block design (RCBD) with three replications in the 2019 early cropping season using standard field management practices at the experimental fields of Guangdong Academy of Agricultural Sciences (GDAAS), Guangzhou city, Guangdong Province, China. Additionally, there were numerous studies have adopted RCBD with three or two replications in one environment [1-6]. Therefore, we firmly believe that our experimental design ensures the accuracy and reliability of experimental data.
Point 2: Furthermore, there are is no information on the experimental trial. Please state the location and year the experiment was planted. What was the row length and plant density? What was the unit of evaluation, how many rows did you plant per entry?
Response 2: Thank you for your comments. All hybrids and restore lines were evaluated and arranged in a randomized complete block design (RCBD) in triplicate in the 2019 early cropping season using standard field management practices at the experimental fields of Guangdong Academy of Agricultural Sciences (GDAAS), Guangzhou city, Guangdong Province, China. All materials were planted in three rows, with six plants per line in one row. The planting density was 16.7 × 16.7 cm. Please refer to the Materials and Methods section for detailed information on the experimental trail. Thanks.
Point 3: In addition, the authors did not respond to my comments on the amount of money, effort and labor that goes into the development of hybrids. To my knowledge, it is cheaper and more productive to spend this amount of time and money to develop superior inbred lines in rice.
In the case of rice, it is very expensive to develop hybrids because rice is a self pollinated crop with a predominance of additive gene action, meaning that inbred lines can be developed that are as high yielding as hybrid rice. The authors do not mention anything about the significant amount of money, effort and labor that goes into developing rice hybrids as compared to inbred lines. Please add a separate section addressing these important concerns and why you believe that developing hybrids is a cost-effective solution to rice breeding.
Response 3: General combining ability (GCA) reflects additive gene actions, is directly related to the breeding value of parents and is theoretically fixable; while specific combining ability (SCA) reflects non-additive gene action resulting from dominance, overdominance, and epistatic effects, which is associated with the performance of hybrids and is non-fixable. In the present study, our results suggested that the performance of studied traits was determined by both additive and non-additive effects, with additive effects dominating, implying that the parent lines exhibited high breeding value with potential heterosis. Compared to inbred lines, screening of commercial value F1 hybrids still requires greater investment. However, the hybrid often exhibits superior performance than its parents, i.e., heterosis or hybrid vigor, which has been widely used in multiple crops that leading to tremendous increase in yield. If the heterosis is predominated by additive gene actions, it is theoretically possible to select homozygous dominant individuals through gene separation and recombination to fix the heterosis to develop superior inbred lines. However, it is unlikely to isolate pure dominant individuals in practice due to genetic linkage, gene interactions, and the fact that many quantitative traits are controlled by minorgene, et al. It should be pointed out that if the magnitude of heterosis is completely determined by the additive effect of favorable dominant genes, that is, it is completely consistent with the dominance hypothesis, then the yield of single hybrid produced between inbred lines cannot exceed the total yield of the two parents. In fact, there are three major hypotheses, dominance, overdominance, and epistasis, have been proposed to explain heterosis. Additionally, according a recent study, the additive and dominant allele-specific expression genes (ASEGs) play important roles for heterosis, and the patterns of CG methylation in gene bodies could play crucial roles in heterosis [7]. Noteworthy, the recent results implied that the ratio of differentially methylated regions in CG context in exons to transcription start sites between the parents could be a feasible predictor for heterosis level, as the ratio was significantly negative correlation with the magnitude of heterosis of their hybrids [7]. Therefore, intensive efforts to overall elucidate the molecular mechanism of heterosis remains to be devoted. We have followed your suggestions to add a separate section to addressing your concerns. Please refer to the revised manuscript for detailed information. Thanks.
Point 4: Due to the aforementioned problems, I believe that this paper may be published as a note rather than a regular article.
Response 4: We greatly appreciate your comments on this text for helping us substantially improve the manuscript. We sincerely hope that the revised manuscript will receive your recognition and recommend it to be published as an article. Thanks.
References
- Rahman, M. M.; Sarker, U.; Swapan, M. A. H.; Raihan, M. S.; Oba, S.; Alamri, S.; Siddiqui, M. H., Combining Ability Analysis and Marker-Based Prediction of Heterosis in Yield Reveal Prominent Heterotic Combinations from Diallel Population of Rice. Agronomy 2022, 12, (8), 1797.
- Gaballah, M. M.; Attia, K. A.; Ghoneim, A. M.; Khan, N.; EL-Ezz, A. F.; Yang, B.; Xiao, L.; Ibrahim, E. I.; Al-Doss, A. A., Assessment of Genetic Parameters and Gene Action Associated with Heterosis for Enhancing Yield Characters in Novel Hybrid Rice Parental Lines. Plants 2022, 11, (3), 266.
- Aldaej, M.; El-Malky, M.; Sattar, M.; Rezk, A.; Naqqash, M.; Al-Khayri, J., Rice (Oryza sativa L.) Breeding among Hassawi Landrace and Egyptian Genotypes for Stem Borer (Chilo agamemnon Bles.) Resistance and Related Quantitative Traits. Phyton 2022, 91, 1-18.
- Gramaje, L. V.; Caguiat, J. D.; Enriquez, J. O. S.; dela Cruz, Q. D.; Millas, R. A.; Carampatana, J. E.; Tabanao, D. A. A., Heterosis and combining ability analysis in CMS hybrid rice. Euphytica 2020, 216, 1-22.
- Ghosh, S.; Rastogi, N.; Sarawgi, A.; Chandrakar, P.; Haldar, A., Combining ability analysis using CMS breeding system in rice. ORYZA- An International Journal on Rice 2020, 50, 52-57.
- Kumar, S. S.; Yadav, V. K.; Bhadana, V. P.; Yadav, M. C.; Sundaram, R., Wide Compatibility Gene Approaches and Heterosis Relationship in Japonica x Indica Hybrid Rice (Oryza sativa L.). Molecular Breeding 2016, 7, (11), 1-12.
- Fu, C.; Ma, C.; Zhu, M.; Liu, W.; Ma, X.; Li, J.; Liao, Y.; Liu, D.; Gu, X.; Wang, H., Transcriptomic and methylomic analyses provide insights into the molecular mechanism and prediction of heterosis in rice. The Plant Journal 2023, 115, 139-154.

Reviewer 2 Report (New Reviewer)
Dear authors.
Thank you for handling the queries.
The use of English was improved by the authors.
Author Response
Point 1: Minor editing of English language required
Response 1:Thank you for your comments. We have submitted the manuscript to a professional service Genesis Technology Communication (Beijing), Co., Ltd for language corrections. We really hope that the flow and language level have been substantially improved. Thanks.
Round 3
Reviewer 1 Report (New Reviewer)
I would like to thank the authors for their effort in answering my comments.
As I mentioned before, data on agronomic polygenic traits that are based only in one environment and three replications are not very accurate. I disagree that this experimental design ensures the accuracy and reliability of experimental data since three replications in only one environment are not enough to make valid conclusions. You can only make preliminary conclusions till you apply your tests in other environments as well.
The authors replied that “numerous studies have adopted RCBD with three or two replications in one environments. Therefore, we firmly believe that our experimental design ensures the accuracy and reliability of experimental data.”
I would respectfully like to argue that we are reviewing this research study with the specific objective and not other manuscripts, thus, the argument that 5 studies have adopted RCBD with 3 replications is not good. Comparing this study with other research studies is not the objective. I would think that the objective is to improve the accuracy of data and results in this study.
In addition, the new revisions have errors in English syntax and grammar. Please check the quality of English and revise accordingly.
Please state the month/year you planted and the month/year you harvested the research trial.
I believe that many statisticians and plant breeders will argue that three replications are not enough to make valid conclusions about such an important research topic. I think that adding more replications will improve the quality of this research study and will add significant merit to the conclusions.
I would think that it is a benefit to publish this study as a note. Once the authors get more data, they can publish it as an article by adding the results and polishing the conclusions.
Moderate Editing of some revised text is required. Some sentences do not read well.
Author Response
Point 1: As I mentioned before, data on agronomic polygenic traits that are based only in one environment and three replications are not very accurate. I disagree that this experimental design ensures the accuracy and reliability of experimental data since three replications in only one environment are not enough to make valid conclusions. You can only make preliminary conclusions till you apply your tests in other environments as well.
Response 1: Thank you for your comments. We deeply admire your rigorous and persistent scientific spirit. At present, we are concerned that we cannot perform the tests in other environments in a short time due to objective factors such as the long growth cycle of rice. We’ll conduct further tests in other environments when we have available fields in other environments.
Point 2: The authors replied that “numerous studies have adopted RCBD with three or two replications in one environment. Therefore, we firmly believe that our experimental design ensures the accuracy and reliability of experimental data.”
I would respectfully like to argue that we are reviewing this research study with the specific objective and not other manuscripts, thus, the argument that 5 studies have adopted RCBD with 3 replications is not good. Comparing this study with other research studies is not the objective. I would think that the objective is to improve the accuracy of data and results in this study.
Response 2: Thank you for your review of the issues and objectives of our manuscript. We fully agree with your opinion that the objective of this study is to improve the accuracy of data and results. It should be noted that the objective of citing these articles is to demonstrate that the RCBD with three replications adopted in our present study is feasible to obtain the accuracy and reliability of experimental data, rather than comparing with other research studies. Thanks.
Point 3: In addition, the new revisions have errors in English syntax and grammar. Please check the quality of English and revise accordingly.
Response 3: Thank you for your correction and we are truly for those mistakes. We have carefully checked and revised the errors in main text of syntax and grammar.
Point 4: Please state the month/year you planted and the month/year you harvested the research trial.
Response 4: Thank you for your comments. We have followed your suggestion to add the month/year of the material planting and harvesting in the Materials and Methods section. We stated the dates for sowing and transplanting of the materials, namely “Seeds were sown after completing germination on February 28. Then, thirty days old seed-lings were transplanted in the main field on March 30”. We also state the month/year of the material harvesting as “At maturity, six plants per plot were harvested at 35 days after heading during June to July in 2019 and ……”.
Point 5: I believe that many statisticians and plant breeders will argue that three replications are not enough to make valid conclusions about such an important research topic. I think that adding more replications will improve the quality of this research study and will add significant merit to the conclusions.
I would think that it is a benefit to publish this study as a note. Once the authors get more data, they can publish it as an article by adding the results and polishing the conclusions.
Response 5: Thank you for your comments and suggestions that could greatly improve the quality of our manuscript. However, we could not perform these tests in a short time due to limitation of experimental fields. We’ll conduct further tests in other environments when we have available fields in other environments. With the reference of several studies, we thought the conclusion that raised by RCBD with three replications in field that located near where the cultivars were planted by farmers have valid meanings in rice production. For the meaning of this study on rice production, we sincerely hope that our revised manuscript can be accepted for publish as an article.

This manuscript is a resubmission of an earlier submission. The following is a list of the peer review reports and author responses from that submission.
Round 1
Reviewer 1 Report
Comments to the Authors
The manuscript entitled “Combining ability analysis of yield related traits of elite rice restorer lines Yuenongsimiao and its derived variety” explains the current needs of enhancing the grain yield. However, the study lacks the novelty of the work and conduct of the experiment. The comments are as below.
The authors mentioned that “Meantime, YNSM and YHSM are also be used as elite restorer lines with broad restoring spectrum for hybrid rice breeding, and 67 derived hybrid combinations have been authorized at different levels for more than 110 times in China (from China Rice Data Center). In the work reported here, we identified several different types of restorer-of-fertility (Rf) genes using the corresponding specific markers to revealed the reasons for the broad restoring spectrum of YNSM and YHSM”. In the introduction part of the manuscript, the authors clearly mentioned about the broad restoring spectrum of YNSM and YHSM and also mentioned about the 67 hybrid combinations which have been authorized to cultivate in different levels in China”, after knowing the clear inference from the previous studies, what is the novelty of the study?
The authors claimed the pyramiding of three restorer-of-fertility genes (Rf3, Rf4 and Rf5/Rf1a) in the abstract and four restorer-of-fertility genes (Rf3, Rf4 Rf5/Rf1a and Rf6) in result section. However, the authors could not do any molecular activity in regard to pyramiding or marker-assisted selection except for the screening of the set of restorer and control genotypes using restorer gene based markers.
Nowhere in the manuscript, the authors haven’t mentioned the importance of the targeted restorer genes and their genetic and/or molecular basis linked to combining ability.
The authors evaluated all the 50 combinations along with five restorer in only one season. ANOVA (Table 1) clearly indicated the significant differences among the replications and even the magnitude of replications is more than some of the sources of variation, there may be an error in implementing the experiment or the influence of environment on the experiment. Hence, the conclusions drawn from the experiment not sounds precise and other researcher can’t rely on the results of this experiment.
The authors also claim that “Notably, the results of the GCA effect of YNSM and YHSM on PH indicated that YHSM and YNSM can effectively enhance lodging resistance through reducing the plant height of the derived hybrid combinations”, however they didn’t discuss anywhere in the manuscript regarding the relevance of plant height and GCA effect of restorer genes or any genetic and/or molecular basis of the concept.
The pedigree details of the parental lines used in the study needs to be provided.
The gel pictures need to be labelled properly with amplicon size.
The grammatical corrections, flow of the content and discussion needs to be drastically improved.
Reviewer 2 Report
The manuscript is well-written and provides valuable information on hybrid rice breeding programs. Overall, the Materials and Methods section provides a suitable overview of the experimental design and methods used in the study, but some additional information and explanation would improve the reader's understanding of the procedures and results.
Reviewer 3 Report
The manuscript entitled on “Combining ability analysis of yield related traits of elite rice restorer lines Yuenongsimiao and its derived variety” has been reported by Authors.
Overall the research is good. All the figures and tables are adequate. This paper is of interest, but few important corrections are needed to improve the manuscript quality.
1. Language editing is essential.
2. Reframe the keywords.
3. Figure 1 and 2 not clear.
4. What is the novelty of present research work?
5. How many replications were maintained?
6. I hope few authors’ names were missed (Zhiqiang Fang 1 , Xiuying He 1 and *) check it.
7. What is the outcome of your research? Highlights in conclusion.
8. Comparative studies are very essential. So compare your findings to previous research.